# Bayesian Measurement of Diagnostic Accuracy of the RT-PCR Test for COVID-19

**Nikhil Padhye**

Cizik School of Nursing, The University of Texas Health Science Center at Houston, Houston, TX 77030, USA; nikhil.s.padhye@uth.tmc.edu

**Abstract:** Reverse transcription polymerase chain reaction (RT-PCR) targeting select genes of the SARS-CoV-2 RNA has been the main diagnostic tool in the global response to the COVID-19 pandemic. It took several months after the development of these molecular tests to assess their diagnostic performance in the population. The objective of this study is to demonstrate that it was possible to measure the diagnostic accuracy of the RT-PCR test at an early stage of the pandemic despite the absence of a gold standard. The study design is a secondary analysis of published data on 1014 patients in Wuhan, China, of whom 59.3% tested positive for COVID-19 in RT-PCR tests and 87.6% tested positive in chest computerized tomography (CT) exams. Previously ignored expert opinions in the form of verbal probability classifications of patients with conflicting test results have been utilized here to derive the informative prior distribution of the infected proportion. A Bayesian implementation of the Dawid-Skene model, typically used in the context of crowd-sourced data, was used to reconstruct the sensitivity and specificity of the diagnostic tests without the need for specifying a gold standard. The sensitivity of the RT-PCR diagnostic test developed by China CDC was estimated to be 0.707 (95% Cr I: 0.664, 0.753), while the specificity was 0.861 (95% Cr I: 0.781, 0.956). In contrast, chest CT was found to have high sensitivity (95% Cr I: 0.969, 1.000) but low specificity (95% Cr I: 0.477, 0.742). This estimate is similar to estimates that were found later in studies designed specifically for measuring the diagnostic performance of the RT-PCR test. The developed methods could be applied to assess diagnostic accuracy of new variants of SARS-CoV-2 in the future.

**Keywords:** bayesian modeling; Hamiltonian Monte Carlo; diagnostic uncertainty; expert opinion data; verbal probability

## 1. Introduction

The cause of a disease outbreak that was discovered in Wuhan, China in the last quarter of year 2019 was later identified as a novel coronavirus, labeled SARS-CoV-2 while the corresponding respiratory disease was named COVID-19 [1]. The publication of the SARS-CoV-2 genome [2] was followed by rapid development of reverse transcription polymerase chain reaction (RT-PCR) tests for the diagnosis of COVID-19 that did not cross-react to other known coronaviruses. Among early versions, one test was developed by the National Institute for Viral Disease Control and Prevention (Beijing, China) that targeted the *ORF1ab* and *N* genes of viral RNA while another version was developed in Germany that targeted the *RdRp*, *E*, and *N* genes [3]. RT-PCR tests were developed and implemented thereafter by many laboratories around the world [4–6], even as COVID-19 became a global pandemic that continued to spread rapidly.

The traditional way of measuring the accuracy of any diagnostic test is by comparing its performance to a gold standard that is infallible, or nearly so. The pandemic led to a fast global response in which it was often assumed that RT-PCR tests were the gold standard, partly because better measurement standards were not known. Concern soon mounted in the medical community at the front lines of the pandemic about a suspected high rate of false negatives of the RT-PCR test. These concerns were based on anecdotal evidence, but

they were nonetheless important concerns that received media attention. For instance, in early April 2020, major US news media published editorials warning the public about the problem with false negative RT-PCR tests [7,8].

In a global pandemic, early knowledge can be vital. Moreover, knowledge that is based on data attracts a higher degree of confidence and attention from public health authorities than in knowledge that develops anecdotally. It is with this motivation that this study looks back at data published in late February 2020 to measure the diagnostic accuracy of the RT-PCR test that was used in the early stage of the pandemic in Wuhan. The data were collected in January and February 2020 by Ai et al. [9] in Wuhan, China, with the objective of measuring the accuracy of chest computerized tomography (CT) imaging for diagnosis of COVID-19 in 1014 hospital patients. Although the investigators assumed that RT-PCR was the gold standard, they provided additional information about the status of patients in the form of verbal probabilities of infection (e.g., *highly likely* , *probable*) that were assigned after a review by medical staff. Patients with negative RT-PCR tests but positive chest CT results were deemed to be *highly likely* to be infected if they had clinical symptoms of COVID-19 and disease progression was observed in a repeated chest CT scan. If they had clinical symptoms without progression of the disease in the follow-up chest CT scan, they were judged to have *probable* infection. Thus, in addition to the counts of positives and negatives for chest CT and RT-PCR, there were underutilized data in the form of expert judgments about the patients that are henceforth termed *expert opinion data*.

The novelty of the approach in this study lies in combining the previously described expert opinion data with diagnostic test data and cross-fertilization with methods that were developed in the context of crowd-sourcing for machine learning that opened a path to the measurement of accuracy of the RT-PCR diagnostic test without the need for a gold standard measurement. This approach treats each diagnostic test result as a noisy measurement without the need for one of the test results to be a gold standard. It may not be reasonable to expect that a gold standard test can be developed immediately when confronted with a new disease, such as COVID-19. The present study shows that this limitation need not prevent rigorous estimation of the accuracy of the leading diagnostic test adopted for combating the spread of the disease. This could be of importance not only in a future pandemic, but also in the ongoing COVID-19 pandemic. Several variants of SARS-CoV-2 have been reported and new variants may continue to arise. The need for speedily assessing accuracy of the RT-PCR test for a new variant continues to be important for public health and safety.

## 2. Materials and Methods

### 2.1. Data

The data were described in a study that retrospectively enrolled patients suspected of having COVID-19 who underwent RT-PCR and chest CT imaging diagnostic tests at Tongji Hospital of Tongji Medical College of Huazhong University of Science and Technology in Wuhan, Hubei, China, during a 30-day period in the months of January and February, 2020 [9]. The effective sample size was 1014 and it was reported that 46% were male while the mean age was $51 \pm 15$ years. Throat swab samples were collected and the RT-PCR assays were reported to have used TaqMan One-Step RT-PCR kits from Huirui Biotechnology Co., Ltd. (Shanghai, China), or BioGerm Medical Biotechnology Co., Ltd. (Shanghai, China), both of which were approved for use by China Food and Drug Administration. Chest imaging was done on one of three CT systems at the hospital and two radiologists reviewed the images while being blinded to the molecular test results. The median time interval between the chest CT exams and RT-PCR assays was 1 day.

RT-PCR assays tested positive for 601 patients (59.3%) and negative for the other 413 patients (40.7%). Chest CT exams were positive for 888 patients (87.6%) and negative for the other 126 patients (12.4%). See Table 1 for the joint distribution of the two tests. A large block of 308 patients with conflicting test results were reassessed on the basis of clinical symptoms and serial CT scans. The investigators concluded that 147 of these patients could

be classified as *highly likely* cases of COVID-19 and another 103 could be classified as *probable* cases of COVID-19. Patients in both classifications had clinical symptoms of COVID-19, but repeat CT scans showed progression of disease in the highly likely cases while being stable in the probable cases. In summary, the data include the joint distribution of test results from RT-PCR and chest CT along with expert opinion in the form of verbal probabilities.

**Table 1.** Summary of data for the observed joint distribution of RT-PCR and chest CT test results. Adapted from Ai et al. [9]. See the footnote for a summary of clinical expert opinion data.

|  | Positive Chest CT | Negative Chest CT | Row Sums |
|---|---|---|---|
| Positive RT-PCR | 580 | 21 | 601 |
| Negative RT-PCR | 308 [a] | 105 | 413 |
| Column Sums | 888 | 126 | 1014 |

[a] Expert opinion indicated that COVID-19 was *highly likely* in 147 of these cases and *probable* in another 103 cases.

The described data were selected because the study [9] was done early in the trajectory of the pandemic and because expert opinion data were provided. The motivation for the present study is to show that innovative application of statistical methods can be used to estimate diagnostic accuracy at an early stage of the pandemic despite the absence of a gold standard test.

### 2.2. Statistical Analysis

#### 2.2.1. Diagnostic Accuracy Model

Dawid and Skene [10] developed a maximum likelihood model for data from multiple raters that rigorously accounted for the uncertainty of the true rating of a case. It is commonly used in the context of crowd-sourced data to model the unknown true category from known categorizations by several raters, a common problem in labeling large data sets that are required for deep learning. Their model is applied here in its Bayesian form for estimation of the sensitivity and specificity of diagnostic tests. The RT-PCR and chest CT diagnostic tests are analogous to two raters that independently assign binary ratings to each patient: either they have the disease or they do not. For clarity, it is emphasized that the binary ratings may or may not correspond to the underlying reality, which is that the patient either has the disease or they do not. This underlying reality, which is unknown, is termed *true disease status* of the patient in the rest of this section. The probability of correct assignment depends only on the true disease status of the patient and the accuracy of the test. Although the original model was developed for multiple discrete responses, the simpler case of binary responses is outlined below.

The underlying true disease status for the $i$th patient may be denoted $z_i$ and it can take values binary values 0 and 1. The patients are drawn from a population with disease prevalence $\pi$ and it follows that the true disease status of a patient is expected to follow the Bernoulli distribution:

$$z_i \sim \text{Bernoulli}(\pi). \tag{1}$$

The results of the diagnostic tests carried out on each patient are denoted by $y_{ij}$ which is the binary response of the $j$th diagnostic test to the $i$th patient, sampled from the Bernoulli distribution:

$$y_{ij} \sim \text{Bernoulli}\left(\theta_{j,z[i]}\right), \tag{2}$$

where $\theta_{j,z[i]}$ are model parameters that represent the probability of assigning $y_{ij} = 1$, or COVID-19 positive status, to a patient of true category $z_i$. Thus, $\theta_{j,1}$ is the sensitivity of the $j$th diagnostic test while $\theta_{j,0}$ is its false positive rate. The probability of assigning $y_{ij} = 0$, or COVID-19 negative status, follows automatically since there are only two categories of the response. The specificity is given by $1 - \theta_{j,0}$ while the false negative rate is $1 - \theta_{j,1}$ for the $j$th diagnostic test.

The joint posterior distribution of interest is $p(y, \theta, \pi)$, which factors as

$$p(y, \theta, \pi) = p(y \mid \theta, \pi) \, p(\pi) \, p(\theta). \tag{3}$$

It follows that for known $z_i$ on $N$ patients with two diagnostic tests,

$$p(z, y \mid \theta, \pi) = \prod_{i=1}^{N} \left( \text{Bernoulli}(z_i \mid \pi) \prod_{j=1}^{2} \text{Bernoulli}\left( y_{ij} \mid \theta_{j,z[i]} \right) \right). \tag{4}$$

However, the true disease status $z_i$ is unknown in the absence of a gold standard test for COVID-19. The model is therefore designed to marginalize $z_i$ out of the above equation by summing over both possible values of $z_i$ for each patient:

$$p(y \mid \theta, \pi) = \prod_{i=1}^{N} \sum_{k=0}^{1} \left( \text{Bernoulli}(k \mid \pi) \prod_{j=1}^{2} \text{Bernoulli}\left( y_{ij} \mid \theta_{j,k} \right) \right). \tag{5}$$

The Bayesian sampling was designed to explore the target distribution for the right hand side of Equation (5). The original maximum likelihood solution of Dawid and Skene [10] is equivalent to optimization of $p(y \mid \theta, \pi)$ with uniform priors for the parameters. The prior distributions that will be used here for $\pi$ and $\theta$ are discussed below.

### 2.2.2. Prior for Prevalence

The informative prior distribution of the prevalence $\pi$ of COVID-19 in the patient population was calculated from expert opinions that had been expressed as verbal probabilities. Before outlining the procedure, it is important to note that we only need to estimate the prevalence of COVID-19 in the population of 1014 patients recruited in the study, not in the population of Wuhan or any other geographical region of China.

The ICD 203 report [11] provides a mapping from the expert opinions expressed in the format of verbal probabilities to quantitative probabilities. Probability range for *probable* is $(0.55, 0.80)$ while the range for *highly likely* is $(0.80, 0.95)$, where equivalence was assumed between the terms *highly likely* and *very likely*. Support for validity of the ICD 203 report [11] comes from a recent study [12] carried out in the framework of fuzzy membership functions to account for the linguistic uncertainty of verbal probability terms. The authors reported that their findings were in alignment with the ICD 203 report. As another check, when we used membership functions reported by Wintle et al. [12] to calculate the centroids for verbal probabilities in the fuzzy systems approach [13], they were found to be 0.75 for *probable* and 0.84 for *highly likely*, which are contained within the corresponding ranges specified by the ICD 203 report.

The prior distribution $\pi$ was calculated for two scenarios. In the first scenario, the verbal probability distributions for *probable* and *highly likely* were beta distributions with their 0.005 and 0.995 quantiles set equal to the probability ranges of the ICD 203 report. That is, 99% of the area under the curve of the beta distributions was contained in the ranges identified in the ICD 203 report. In the second scenario, the verbal probability distributions for *probable* and *highly likely* were assumed to be uniform distributions on the ranges $(0.55, 0.80)$ and $(0.80, 0.95)$, respectively, and zero outside that range. In each scenario, the first step was the calculation of the joint probability distribution of 103 *probable* and 147 *highly likely* cases for the set of patients for whom verbal probabilities had been assigned by experts due to uncertainty about created by conflicting diagnostic tests. In the second step, the information from the unambiguous cases was added, resulting in the exact prior distribution, $\pi$, for both scenarios. Lastly, simpler beta distribution fits were found to each of the two exact prior distributions by minimization of the Kullback-Liebler divergence. This was necessary to avoid a major slowdown of the Bayesian computation caused by the inability to express the exact prior with built-in functions in *Stan*. The

resulting distributions are shown in Figure 1. Appendix A contains details of the calculation of the prior distribution.

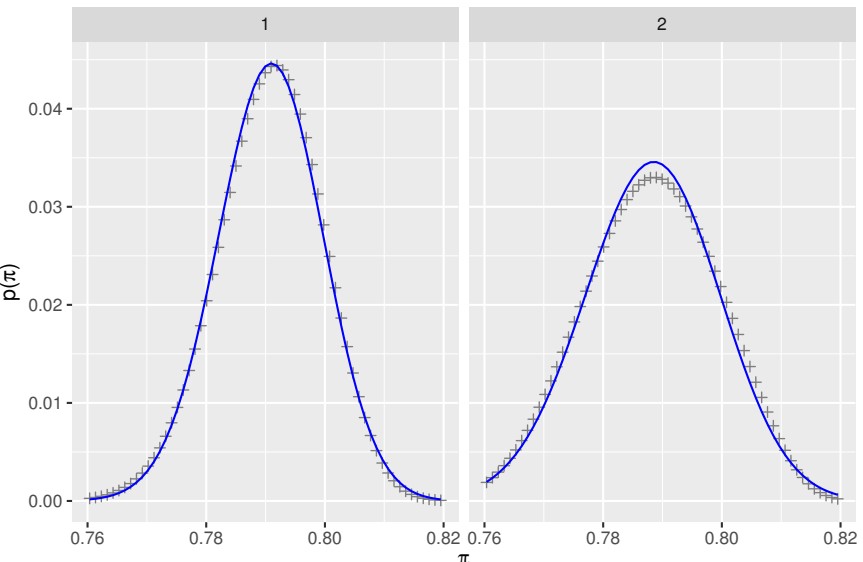

**Figure 1.** Prior distributions for prevalence of COVID-19 under two sets of assumptions that are described in Section 2.2.2. Each panel shows the exact prior (+ symbols) calculated by numerical integration and the beta distribution (blue line) fitted by minimizing KL divergence. The range of prevalence, $\pi$, has been shortened to bring out the differences while still including the central 95% interval.

### 2.2.3. Priors for Sensitivity and False Positive Rate

Non-informative prior distributions were used for the sensitivity of both tests: $p(\theta_{j,1}) = Beta(0.5, 0.5)$ for $j = 1, 2$. Weakly informative prior distributions were used for the false positive rates in order to assist with model convergence and to avoid inappropriate inferences. The 0.005 and 0.995 quantiles of $p(\theta_{j,0}) = Beta(2.03, 10.33)$ are located at false positive rates of 0.01 and 0.50, respectively. Priors are shown in Figure 2.

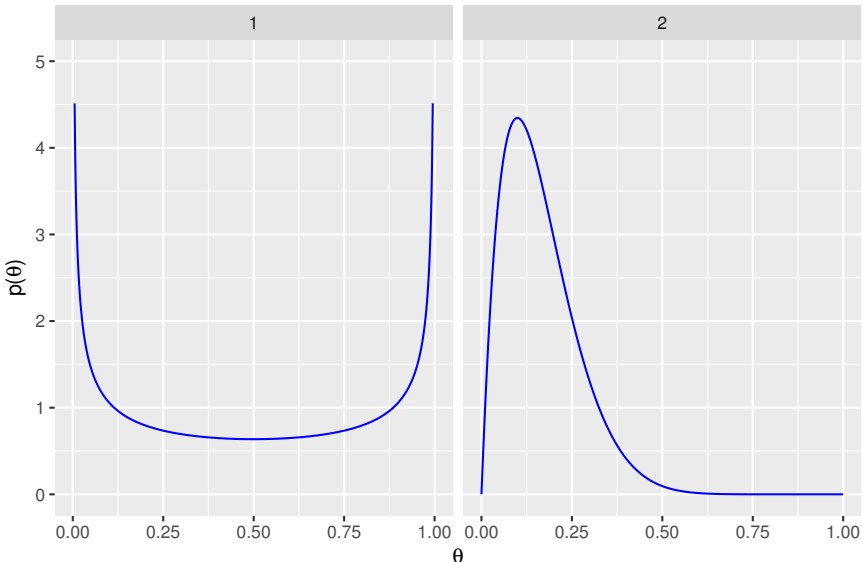

**Figure 2.** Prior distributions for $\theta$ parameters are shown. Panel 1 displays the non-informative prior for sensitivity, $p(\theta_{j,1}) = Beta(0.5, 0.5)$ for $j = 1, 2$. Panel 2 displays the weakly informative prior for false positive rate, $p(\theta_{j,0}) = Beta(2.03, 10.33)$.

### 2.2.4. Implementation in Stan

*No U-turn sampling* (NUTS) was implemented in *Stan* with the package *RStan* [14] that provides an interface between *Stan* and *R*. *Stan* uses Hamiltonian Monte Carlo sampling, which was designed in this application to calculate the target distribution of logarithm of the probability distribution specified in Equation (5). The *Stan* code used simplexes for parameters $\pi$ and $\theta_{j,k}$ to ensure that probabilities of categorization into 0 and 1 levels always sum to 1. The priors were specified as Dirichlet distributions, which reduce to beta distributions for binary categorization. Ten chains were used while running the models in *RStan*, with 5000 iterations per chain that were run in parallel on 20 cores of an Intel i7 processor. Marginalizing out the true status $z_i$ from Equation (4) was essential since *Stan* is not designed for estimation of discrete parameters. Although the discrete parameter was marginalized, estimates were obtained of the probabilities of the true diagnosis, $p(z_i \mid \theta, \pi)$. Statistical analysis was done using the R programming language [15] in the RStudio software environment [16].

### 3. Results

Estimates of diagnostic accuracy of RT-PCR and chest CT diagnostic tests are shown in Table 2 and Figure 3. The table includes estimates from both sets of priors for the prevalence of COVID-19. One of these arose from fitting beta distributions to ranges of probability associated with verbal probability terms (*prior 1*) while the other arose from assuming uniform distributions of verbal probabilities (*prior 2*). The estimated parameters were not markedly different for these priors. Unsurprisingly, the 95% posterior intervals (or credible intervals) were slightly wider for the second prior that arose from uniformly distributed verbal probabilities. Although there were warnings about divergence (10 and 11 divergences for the two models), further inspection showed no patterns in the location of divergences. Moreover, the Markov chains mixed well and resulted in stable estimates. The maximum value of the potential scale reduction statistic was 1.002, well within the desired range $< 1.01$. The estimated sample sizes (ESS) exceeded the recommended value of 100 times the number of chains.

**Table 2.** Estimated sensitivity and specificity of RT-PCR and chest CT diagnostic tests for COVID-19. Max value of the potential scale reduction statistic was 1.002, indicating convergence of Markov chains for all parameters shown in the table.

| Test | Parameter | Prior [a] | Mean | SD | 95% Posterior Int. | | ESS [b] |
|------|-----------|-----------|------|-----|------|------|---------|
| | | | | | Lower | Upper | |
| RT-PCR | Sensitivity | Prior 1 | 0.706 | 0.022 | 0.665 | 0.750 | 8807 |
| | | Prior 2 | 0.707 | 0.023 | 0.664 | 0.753 | 6828 |
| | False Negative Rate | Prior 1 | 0.294 | 0.022 | 0.250 | 0.335 | 8807 |
| | | Prior 2 | 0.293 | 0.023 | 0.247 | 0.336 | 6828 |
| | Specificity | Prior 1 | 0.859 | 0.042 | 0.781 | 0.949 | 7475 |
| | | Prior 2 | 0.861 | 0.043 | 0.781 | 0.956 | 5591 |
| | False Positive Rate | Prior 1 | 0.141 | 0.042 | 0.051 | 0.219 | 7475 |
| | | Prior 2 | 0.139 | 0.043 | 0.044 | 0.219 | 5591 |
| Chest CT | Sensitivity | Prior 1 | 0.992 | 0.009 | 0.970 | 1.000 | 6518 |
| | | Prior 2 | 0.992 | 0.009 | 0.969 | 1.000 | 5355 |
| | False Negative Rate | Prior 1 | 0.008 | 0.009 | 0.000 | 0.030 | 6518 |
| | | Prior 2 | 0.008 | 0.009 | 0.000 | 0.031 | 5355 |
| | Specificity | Prior 1 | 0.610 | 0.063 | 0.488 | 0.736 | 8473 |
| | | Prior 2 | 0.607 | 0.067 | 0.477 | 0.742 | 6403 |
| | False Positive Rate | Prior 1 | 0.390 | 0.063 | 0.264 | 0.512 | 8473 |
| | | Prior 2 | 0.393 | 0.067 | 0.258 | 0.523 | 6403 |

[a] Priors 1 and 2 differ in modeling verbal probability—see Section 2.2.2; [b] Effective sample size after accounting for autocorrelated samples.

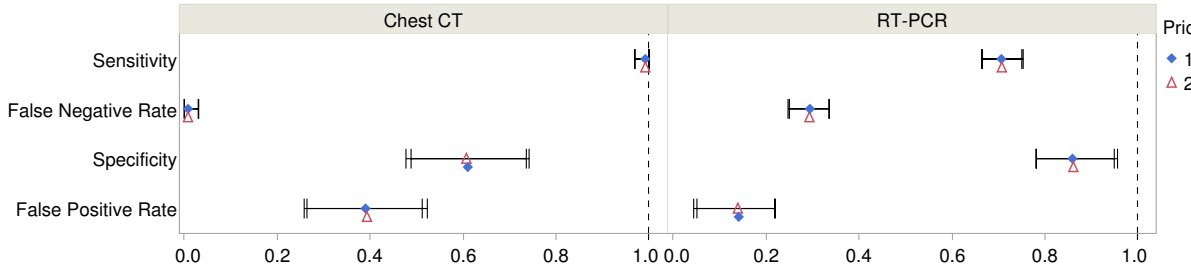

**Figure 3.** Estimated values of diagnostic accuracy parameters and their 95% posterior intervals are shown for chest CT and RT-PCR for two choices of Bayesian priors. For numerical values, see Table 2.

The posterior estimate for the prevalence proportion, $\pi$, was 0.793 (95% Cr I: 0.770, 0.815). The models also provided estimates of the probability of having COVID-19 for each patient and these are shown in Table 3 for each of the four combinations of RT-PCR and chest CT test results. The values ranged from a low of 0.017 (95% Cr I: 0.000, 0.063) when both tests were negative to a high of 0.980 (95% Cr I: 0.966, 0.993) for two positive tests. In the scenario of conflicting test results, the probability of infection was higher for a positive chest CT exam than for a positive RT-PCR test. However, there was large uncertainty in the probability of having COVID-19 when chest CT was negative but RT-PCR was positive. This may not be surprising since Table 1 shows that there were only 21 patients with this combination of diagnostic test results.

**Table 3.** Estimated probabilities of COVID-19 for combinations of RT-PCR and chest CT diagnostic test results. Estimates are from the second model based on *prior 2*, explained in Section 2.2.2. Max value of the potential scale reduction statistic was 1.003, indicating convergence of Markov chains for each estimate.

| RT-PCR | Chest CT | Mean | SD | 95% Posterior Int. | | ESS [a] |
| | | | | Lower | Upper | |
|---|---|---|---|---|---|---|
| Negative | Negative | 0.017 | 0.018 | 0.000 | 0.063 | 5985 |
| | Positive | 0.766 | 0.052 | 0.655 | 0.860 | 4863 |
| Positive | Negative | 0.208 | 0.219 | 0.000 | 0.768 | 4169 |
| | Positive | 0.980 | 0.007 | 0.966 | 0.993 | 8052 |

[a] Effective sample size after accounting for autocorrelated samples.

Post-test predictive values of a test depend on the prevalence of the disease, or on the expected pre-test probability of the disease for any patient. Under conditions of low prevalence, a positive RT-PCR test can greatly increase the post-test probability of presence of COVID-19 infection in the patient. Conversely, a negative test result lowers the post-test probability of a COVID-19 infection. However, the high rate of false negatives for the RT-PCR test have resulted in a diminished magnitude of the change in probability, as may be seen in Figure 4. For instance, if a patient is deemed to have a 50% pre-test chance of having COVID-19 infection, a subsequent negative result from RT-PCR would then lower the chance of infection to 25.4%, which is still quite high.

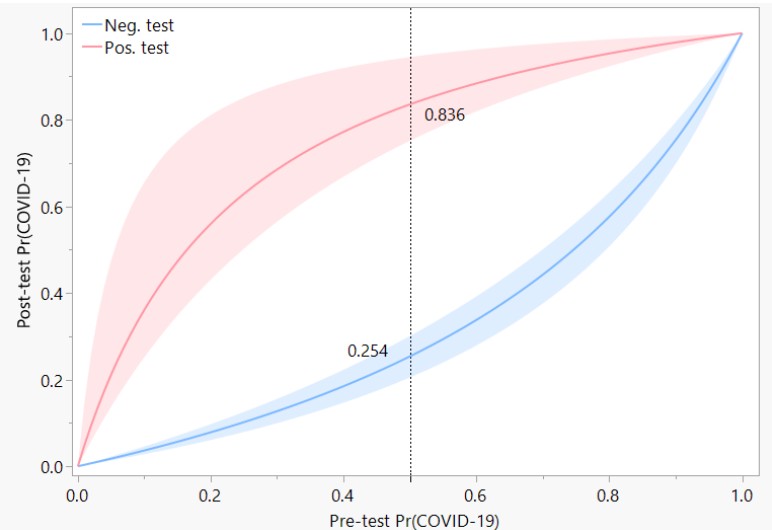

**Figure 4.** Post-test predictive values of the RT-PCR test are shown as a function of the pre-test expectation, which can also be considered to be the prevalence of COVID-19 in the local community. Red and blue curves represent the probability of COVID-19 after a positive or negative test result, respectively, with the width of each band indicative of the 95% posterior interval of the probability. The dotted reference line shows an example of the change in probability from 0.5 pre-test to a post-test value of 0.836 if the RT-PCR test was positive, or to 0.254 if the RT-PCR test was negative.

## 4. Conclusions

RT-PCR tests are commonly used for the diagnosis of many influenza viruses and coronaviruses. These tests are often treated as the gold standard in comparisons made to other diagnostic methods, which has led to a rarity of estimates of their diagnostic accuracy in clinical practice. The virus culture process is considered a better standard, but it takes several days instead of the few hours needed for RT-PCR tests. In one such comparison [17], RT-PCR was found to have sensitivity greater than 96% relative to virus culture for the diagnosis of H1N1 influenza. Similarly, high accuracy of RT-PCR has also been reported for the MERS coronavirus [18]. On the other hand, low accuracy has been reported for detection of SARS coronavirus with real-time RT-PCR [19,20], although rates of detection were improved with the refinement of laboratory methods [21]. These variable estimates show that diagnostic accuracy of RT-PCR tests cannot be automatically assumed to be the gold standard for each new virus or variant of the coronavirus.

In the COVID-19 pandemic, rapid development of RT-PCR diagnostic tests that targeted the detection of different genes from the viral RNA has been a critical element of global response to the public health crisis. Laboratory testing has shown that RT-PCR assays can detect viral loads as small as 3.2 RNA copies per reaction [3] and that it does not cross-react to other known coronaviruses, particularly when the primer for the assay is well-chosen [4]. Despite the success in laboratory testing, concerns about how well the tests have worked in practice arose from an early stage, e.g., [7,8]. There are several factors that can impact the clinical accuracy of RT-PCR tests. One factor is the viral distribution by physical location, such as the differences in positive rates of RT-PCR in nasopharyngeal versus oropharyngeal swabs, or in the sputum and bronchoalveolar lavage fluid [22,23]. Other factors include the timing of the test relative to disease onset, severity of the infection, adequacy of the volume of fluids collected in the swab, and deviations from the laboratory-recommended protocol under real-world conditions. In terms of clinical decision-making, any of the causes of failure of the test lead to incorrect diagnoses. Error rates of diagnostic testing using RT-PCR for COVID-19 that were estimated in this study may be considered to have arisen from the cumulative impact of these sources of error.

The main contribution of this study is to demonstrate that it was possible to measure the accuracy of RT-PCR diagnoses in late February 2020 by utilizing published data about

RT-PCR tests, chest CT, and expert opinions about infections. This is important because it provides a roadmap to speed up the measurement of diagnostic accuracy of new diseases in the future. One key novelty was in the utilization of expert opinion data in the form of verbal probabilities in addition to diagnostic data to estimate the prior proportion of COVID-19. Another novelty of the study is in replacing the common reliance on a gold standard test with rigorous accounting of all possible true states of each patient. For this, a modeling approach developed by Dawid and Skene [10] was used, more commonly used in data labeling or annotation for large data sets used in machine learning. Our analysis showed 70.7% sensitivity of the RT-PCR diagnostic test that was used in Wuhan during the early stage of the pandemic, designed by China CDC and implemented with TaqMan One-Step RT-PCR kits. Even at the upper end of the 95% posterior interval the sensitivity reached only 75.3%, which implies that the false negative rate exceeded 24.7% and it could have been as high as 33.6%. The high rate of false negatives is likely to be among the main reasons for the difficulty in controlling the breakout in its early stages.

One way to reduce the impact of the high false negative rate for RT-PCR tests would be through the conduct of sequential tests. For instance, consider a patient population with 50% prevalance of COVID-19, which was certainly not impossible, given that Ai et al. [9] reported a 59.3% positive rate for RT-PCR tests. Negative result from a single RT-PCR test would reduce the 50% pre-test chance of infection in a patient to about half that value, but if a second test were to be administered, the combination of two negative test results would reduce the post-test chance of infection to 10%, approximately.

Comparison of findings of the present study with other studies of the diagnostic accuracy of RT-PCR indicate that the estimates are largely in alignment. Kovács et al. [24] used a reverse calculation that considered the chest CT as gold standard and reported the sensitivity and specificity of the same version of the RT-PCR test to be 65% and 83%, respectively. Two other reverse calculations carried out by the same authors [24] using data from China and Italy [25,26] provided estimates as low as 47% for sensitivity and as high as 100% for specificity. However, the authors acknowledged that the reverse calculations underestimated sensitivity and overestimated specificity of the RT-PCR tests. This may not be surprising because diagnostic accuracy rates obtained by comparison to a gold standard can produce biased estimates and they can artificially limit the performance of the test if the gold standard is flawed [27]. An analysis of data from seven longitudinal studies found that the probability of false negatives of RT-PCR tests reached their lowest value of 20% on the third day after onset of symptoms before starting to increase again [19]. An Italian study of multiple RT-PCR tests that targeted different genes of viral RNA used a repeated testing design in the emergency room and reported sensitivity values ranging from 62% to 94% [28]. Woloshin et al. [29] concluded that after consideration of current evidence, sensitivity and specificity values of 70% and 95% were reasonable estimates for RT-PCR tests.

Among reasons for the low sensitivity of RT-PCR might be that the Wuhan study used throat swabs rather than nasal swabs and that the optimal timing for testing was still in the process of being discovered. For example, a later study of disease propagation among 4950 quarantined Chinese participants reported that the first and second RT-PCR tests on throat-swab samples collected two days apart were positive in 72% and 92% of the 129 people who were eventually diagnosed with COVID-19 [17]. In a study based in Scotland [30] sensitivity of the RT-PCR test was found to increase from 60% for two serial tests to 78% for four serial tests. Using a meta-analysis approach, Tsang et al. [31] reported sensitivity of 68% for throat swabs and 86% for nasal swabs relative to the nasopharyngeal swab as gold standard.

For chest CT, our estimates for high sensitivity (96.9% to 100%) and low specificity (47.7% to 74.2%) were also of comparable magnitude to other reports. A study in Italy by Caruso et al. [25] used RT-PCR as gold standard and reported 97% sensitivity and 56% specificity. Ai et al. [9] reported 96.5% sensitivity and 25.4% specificity in their study in Wuhan which was also based on assuming that RT-PCR was the gold standard test. We

found a substantially higher estimate of specificity based on the same data, which can be attributed to the advantage of taking expert opinion into account and using an analytical method that did not require comparison to a gold standard.

Future directions suggested by this study include the application of the methods to speedily assess accuracy of diagnostic tests for new variants of SARS-CoV-2. Omicron and delta variants arose later in the pandemic and became the dominant strains. As each variant is discovered, it is hoped that the variant is not so distinct that it might evade detection by previously developed diagnostic tests. However, there is no guarantee that this will continue to be the case for new variants. It may be necessary to measure the efficacy of existing or new diagnostic tests for detection of new variants in the absence of a gold standard test, following the novel method described in this study.

Among limitations of this study, the primary one is that the estimated sensitivity and specificity apply to the particular version of the RT-PCR test that was urgently created by China CDC that was being used in Wuhan, China, during January and February, 2020. Although the study findings reported here are consistent with a meta-analysis based on global data [31], one systematic review found a high level of heterogeneity in global data for false negative rates [32], which serves as a warning against indiscriminate generalization of study results to other versions of RT-PCR tests. Another limitation of the study is that the estimation of the prior prevalence could have been improved if expert opinions had been available for all 1014 patients rather than being restricted to a subset of patients with conflicting diagnostic results from RT-PCR and chest CT. Diagnoses also depend on the number of days since infection, which were not available, and therefore ignored. The accuracy rates estimated here may be considered as averages around the time of development of symptoms. If data about the severity of infections and measures of viral load, such as cycle threshold of RT-PCR assays, had been available it might have been possible to explore whether it was the milder cases that tended to be misdiagnosed.

**Funding:** This research received no external funding.

**Institutional Review Board Statement:** Ethical review and approval were waived for this study because no data were collected and secondary analysis was done on published aggregated data.

**Informed Consent Statement:** Patient consent was waived because no data were collected and published aggregated data were used without access to patient-level data.

**Data Availability Statement:** Not applicable.

**Acknowledgments:** The author is grateful to Stanley Cron for providing feedback on the first draft of this manuscript.

**Conflicts of Interest:** The author declares no conflict of interest.

## Abbreviations

The following abbreviations are used in this manuscript:

| | |
|---|---|
| RT-PCR | Reverse transcription polymerase chain reaction |
| COVID-19 | Coronavirus disease 2019 |
| CT | Computerized tomography |
| ICD | Intelligence Community Directive |
| NUTS | No U-turn sampling |

## Appendix A

This section contains details about the calculation of the informative prior distribution of prevalence of COVID-19 in the 1014 patients.

First, we consider the set of patients for whom verbal probabilities had been assigned by experts due to uncertainty about created by conflicting diagnostic tests. The distribution of integer counts of COVID-19 on the set of 250 patients, $\mathcal{S}$, that were assigned verbal probabilities

can be expressed as a joint distribution arising from the distributions over 103 *probable* COVID-19 cases in partition $\mathcal{S}_1$ and 147 *highly likely* COVID-19 cases in partition $\mathcal{S}_2$:

$$p_{\mathcal{S}}(n) = \sum_{m_1,m_2} \iint_{\phi_1,\phi_2=0}^{1} \text{Binomial}(m_1 \mid M_1, \phi_1) \, \text{Binomial}(m_2 \mid M_2, \phi_2) \, f_1(\phi_1) \, f_2(\phi_2) \, d\phi_1 \, d\phi_2 \tag{A1}$$

where the sum is over all non-negative $m_1$ and $m_2$ that satisfy $m_1 + m_2 = n$. Further, $m_1$ cannot exceed $M_1 = 103$, the sample size of $\mathcal{S}_1$, while $m_2$ cannot exceed $M_2 = 147$, the sample size of $\mathcal{S}_2$. The probabilities of being a COVID-19 case are denoted by parameters $\phi_1$ and $\phi_2$ on the two subsets. Verbal probability distributions are denoted $f_1(\phi_1)$ for *probable* and $f_2(\phi_2)$ for *highly likely* COVID-19 cases.

The joint distribution $p_{\mathcal{S}}(n)$ was calculated with numerical integration for two sets of assumptions. In the first set, the verbal probability distributions $f_1(\phi_1)$ and $f_2(\phi_2)$ were beta distributions with their 0.005 and 0.995 quantiles set equal to the probability ranges of the ICD 203 report. The beta distribution was selected because it is conjugate to the binomial distribution. This resulted in $f_1(\phi_1) = Beta(60.84, 28.29)$ and $f_2(\phi_2) = Beta(101.95, 12.92)$, where the shape parameters for $Beta(\alpha, \beta)$ were estimated by matching quantiles using the *Beta.Parms.from.Quantiles* script [33] written in the *R* programming language. The densities peaked at $\phi_1 = 0.69$ and $\phi_2 = 0.89$. In the second set, the verbal probability distributions $f_1(\phi_1)$ and $f_2(\phi_2)$ were assumed to be uniform distributions on the ranges $(0.55, 0.80)$ and $(0.80, 0.95)$.

Second, addition of the unambiguous COVID-19 cases and converting the count to a proportion, $\pi$, of the 1014 patients yielded the desired informative prior distribution, $p(\pi)$. For the case of beta distributions of verbal probabilities, $p(\pi)$ peaked at $\pi = 0.792$, with 0.025 and 0.975 quantiles equal to 0.771 and 0.807, respectively. For uniformly distributed verbal probabilities, $p(\pi)$ peaked at $\pi = 0.789$, with 0.025 and 0.975 quantiles located at 0.765 and 0.810, respectively. Thus, approximately 79% of the patient population at the hospital was infected and it is not surprising that the assumption of uniform distributions for verbal probabilities resulted in more uncertainty of the prevalence, reflected in its wider 95% confidence interval relative to the assumption of beta distributions for verbal probabilities.

Lastly, simpler fits were found to each of the two exact prior distributions that were described above. This was necessary for speed of the Bayesian implementation since calculation of the exact prior is slow and a closed-form expression is not available for it. Minimization of the Kullback-Liebler (KL) divergence $D_{KL}(\hat{p}(\pi) \parallel p(\pi))$ resulted in an estimate of $\hat{p}(\pi) = Beta(1682.88, 445.42)$ for the first case that started with beta distributions for verbal probabilities. The minimum divergence was $1.46 \times 10^{-3}$, which is less than the KL divergence between a *t* distribution with $df = 33$ and the standard normal distribution, for example. The modes matched within 0.001 and the largest difference in the 95% critical values was less than 0.002. In the second case that started with uniform distributions for verbal probabilities, the estimate was found to be $\hat{p}(\pi) = Beta(1016.15, 273.25)$. The minimum KL divergence was $3.29 \times 10^{-3}$, with match tolerances for the peak and 95% critical values being less than $4.30 \times 10^{-3}$ and $1.07 \times 10^{-3}$, respectively.

KL divergences were calculated using the *flexmix* package [34] and optimized with the Nelder-Mead method [35] in *R*.

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
