# Peer review of "Bayesian Measurement of Diagnostic Accuracy of the RT-PCR Test for COVID-19"

_2673-8244, doi:10.3390/metrology2040025_

Round 1

Reviewer 1 Report

1.     In what ways can the manuscript be retrospective when it covers a period of time equal to +/-1 year?

2. The title needs to be very accurate. Here it needs an information in what place have the research been taken place. 

3.     According to my reviewer’s knowledge, each scientific paper needs to be divided into rigidly established rules, which is: an abstract, an introduction, methodology (materials and methods), results, conclusions/recommendations. According to that, the section Methods should be extended to: Materials and methods (line: 62).

4.     Quotations in lines 31, 33, 35 are missing.

5.     What does “verbal probability data” mean? (lines: 52-53).

6.     What does “a true diagnosis” mean? (line: 99).

7.     What does “a true status” mean (line: after 105).

8.     What is the basic foundation for the research: methodological, organizational, etc.?

9.     Why these kind of methods have been used?

10.     What are the restrictions of these methods?

11.  What are the criteria for selecting these methods?

12.   What are the criteria for selecting the research population, including the cited literature as a background for the research.

13.  Let’s try to transform conclusions directly resulting from the analyses of statistical methods into conclusions of an applcative nature.

14.  What are the first and second order conclusions of the analyses carried out?

15.  Where, in the conclusion, is it proven that hypotheses have been confirmed or rejected?

16.  There are no practical recommendations what to do to eliminate false positive rates?

17.  The literature should be enrichened, especially by the literature which can upgrade the manuscript's title (retrospection).

18.  The sentence in line: 59, and in line: 269 has to be built in a proper grammatical way.

19. It is a good piece of work! It is an interesting topic.

20. The part in lines: 249-254) is crucial and needs to be extended!

21. Take a chance to improve the manuscript!

Reviewer 2 Report

The study entitled "Retrospective measurement of diagnostic accuracy of the RT-PCR test for COVID-19" is of significant importance due to consequences associated with the COVID-19 pandemic. Moreover, the emergence of the Omicron variant and its subvariants put the diagnostic procedures into the dilemma and doubts. I appreciate the efforts behind the present work. 

I have read the article with great interest and it is such an impressive work. However, I wonder if there is any chance the results can be presented in the form of any figure/figures.

Additionally, I suggest to incorporate the conclusion section which highlights the major findings of the present article.

Furthermore, the future directions should be provided while mentioning how this study can be of great significance amid the evolution of SARS-CoV-2 and the generation of various variants of SARS-CoV-2. The above changes will increase the reach of the manuscript to the readers. 

Best Wishes 

Round 2

Reviewer 1 Report

Dear Author,

I appreciate your work done to improve the quality and readibility of the manuscript. I see a lot of goods in it!

Nevertheless, please modify following things:

a) what does trude disease mean? (line 108),

b) what does "an expert opinion" mean? (line: 54, 94-95), in what way has is been achieved?,

c) under consideration and under conditions (line 108) - these wording should be changed to another word.

I recommend to amend the changes before letting the manuscript to be published.

Best wishes

your Reviewer
